# Methods and Limits for Micro Scale Blood Vessel Flow Imaging in Scattering Media by Optical Feedback Interferometry: Application to Human Skin

**DOI:** 10.3390/s21041300

**Published:** 2021-02-11

**Authors:** Adam Quotb, Reza Atashkhooei, Simone Magaletti, Francis Jayat, Clement Tronche, Julien Goechnahts, Julien Perchoux

**Affiliations:** 1LAAS-CNRS, Université de Toulouse, CNRS, INP-ENSEEIHT, 31400 Toulouse, France; magaletti.simone@gmail.com (S.M.); Francis.jayat@laas.fr (F.J.); clement.tronche@laas.fr (C.T.); julien.goechnahts@enseeiht.fr (J.G.); julien.perchoux@laas.fr (J.P.); 2Centre for the Development of Sensors, Instruments and Systems, Universitat Politècnica deCatalunya (UPC-CD6), Rambla Sant Nebridi, 10, E08222 Terrassa, Spain; atashkhooei@gmail.com

**Keywords:** micro scale blood flow, imaging, optical feedback interferometry

## Abstract

At the micrometric scale, vessels or skin capillaries network architecture can provide useful information for human health management. In this paper, from simulation to in vitro, we investigate some limits and interests of optical feedback interferometry (OFI) for blood flow imaging of skin vascularization. In order to analyze the tissue scattering effect on OFI performances, a series of skin-tissue simulating optical phantoms have been designed, fabricated and characterized. The horizontal (2D) and vertical (depth penetration) sensing resolution of the OFI sensor have been estimated. The experimental results that we present on this study are showing a very good accordance with theoretical models. In the case of a skin phantom of 0.5 mm depth with a scattering coefficient from 0 to 10.8 mm${}^{-1}$, the presented OFI system is able to distinguish a pair of micro fluidic channels (100 µm × 100 µm) spaced by 10 µm. Eventually, an in vivo test on human skin is presented and, for the first time using an OFI sensor, a 2D blood flow image of a vein located just beneath the skin is computed.

## 1. Introduction

Vessels or skin capillaries network architecture study can provide useful information for human health management [1]. In the case of skin cancer pathology, the presence of a malign tumor involves an atypical vascular pattern. In that situation the analysis of the tumor angiogenesis is a good indicator for practitioners diagnosis [2].

Laser Doppler flowmeters (LDF) are non invasive sensors that are able to measure micro circulation of biological tissue. Optical fiber sensing probe [3] and free space sensors [4] are commonly accepted methods for continuous and long-term monitoring of blood flow just under the skin.

In this application domain, Optical Feedback Interferometry (OFI) or Self Mixing Interferometry (SMI) has been introduced in the last decade [5]. The main advantage of OFI sensors is the use of a laser as the light source, the interfering medium and the coherent detector all in one, making the setup extremely compact, economic and self-aligned, while keeping it contactless and achieving a resolution and accuracy that are comparable to that of classical optical interferometry, as far as it is a coherent detection method. OFI has been proposed for fluid flow measurements in biomedical applications [6] and Zakian et al. performed imaging of a flow channel through a tissue phantom [7].

From simulation to in vivo operation, this paper evaluates the characteristics and parameters of OFI performance for subcutaneaous blood flow measurements. The OFI spatial sensing resolution is evaluated on a dedicated microfluidic chip with skin phantoms presenting different scattering and absorption coefficients. A novel method for blood perfusion computation using the zeroth moment is proposed and investigated. The maximum sensing depth is investigated in order to determine the sensor limits for skin vascularization monitoring. Eventually, an in vivo test is presented demonstrating imaging of a vein on a human forearm.

In Section 2, the effect of scattering media on OFI based flow sensing is theoretically analyzed. Section 3 presents skin phantoms fabrication procedure and characterization method. In Section 4, results regarding simulations and experimental validations of OFI sensing are analysed. The 2D resolution and the maximum sensing depth estimation of our sensor is investigated. In order to illustrate the sensor ability, OFI in vivo blood flow detection tests performed on human volunteers are discussed.

## 2. Influence of Scattering Media on OFI Sensing

The OFI phenomenon is the result of the interference between the light wave propagating inside the laser cavity and the reinjected back-scattered light from a distant target. This optical feedback produces a modulation of the laser emission power which can be observed through the in-package monitoring photodiode. Any disturbance of the laser beam on the optical path, such as scattering, absorption or else is then impacting the amplified photodetected current. Figure 1 sketches the configuration where an OFI sensor is beaming onto a remote target as the laser light travels through a scattering medium placed in the optical path beam between the target and the OFI sensor.

Considering the emitting power $P_{0}$, and the unscatterred transmission coefficient (which is the portion of the laser power being transmitted and not scattered) *T*, the power of the beam after propagation through the medium is $TP_{0}$. Thereby, the reflected power from the target is $RTP_{0}$, where *R* represents the ratio of back-reflected power that would actually re-enter the laser cavity if there were no scattering medium in the path. Then, travelling a second time through the media, it is subject to another attenuation by a factor *T* and eventually the re-injected power $P_{r}$ writes
(1)$$P_{r}=TRTP_{0}=P_{0}RT^{2}.$$

The unscattered transmission coefficient of a multiple scattering media is given by (Equation 2) [8]
(2)$$T=e^{-L(\mu_{t})}=e^{-L(\mu_{a}+\mu_{s})},$$
where *L* is the thickness of the medium, $\mu_{t}$ is the attenuation coefficient which is the sum of the absorption coefficient ($\mu_{a}$) and the scattering coefficient ($\mu_{s}$). Combining Equations (Equation 1) and (Equation 2) results in
(3)$$\frac{P_{r}}{P_{0}*R}=T^{2}=e^{-2L(\mu_{a}+\mu_{s})}.$$

Hence, the re-injected power is proportional to the square of the unscattered transmission.

The sensor signal can be determined by solving Lang and Kobayashi equations [9]. In the case of a unique translating target, a well established formulation is
(4)$$P_{F}=P_{0}\left[1+m\cos(2\pi f_{D}t+\Phi_{D})\right],$$
where $P_{F}$ is the actual power of the laser, $P_{0}$ is the power of the standalone laser, $f_{D}$ is the Doppler frequency shift of the back-scattered wave, $\Phi_{D}$ is a constant phase term and *m* is the modulation index as expressed in (Equation 5):(5)$$m=4\sqrt{\frac{P_{r}}{P_{0}}}\frac{1-R_{2}}{\sqrt{R_{2}}}.$$
In the case of multiple scatterers such as red blood cells in blood flow monitoring, it has been demonstrated [10] that each contribution sums into the laser power expression
(6)$$P_{F}=P_{0}\left[1+\sum_{i}m_{i}\cos\left(2\pi f_{D,i}t+\Phi_{D,i}\right)\right].$$

The electrical signal is the amplification of the photodiode current which can be written as
(7)$$i=I_{0}\sum_{i}m_{i}\cos\left(2\pi f_{D,i}t+\Phi_{D,i}\right).$$

It can be seen that the power of the OFI signal is proportional to $I_{\mathrm{rms}}^{2}$ is then proportional to $m^{2}$ in the case of a single target and ${\left(\sum_{i}m_{i}\right)}^{2}$ in case of multiple contribution. Thus, it means that the signal power is proportional to the ratio $P_{r}/P_{0}$. In order to evaluate the signal power in the case of multiple contributions we have integrated its power spectral density over the frequency as originally proposed by de Mul [11] as the zeroth order moment $M_{0}$ of the signal spectrum (Equation 8).
(8)$$M^{0}=\int_{f_{a}}^{f_{b}}\left(P\left(f\right)-P_{N}\left(f\right)\right)df$$
where *f* is the frequency, and P(f) is the power spectrum density obtained by square module of laser output power FFT. $P_{N}\left(f\right)$ is the power spectrum density when there is no Doppler frequency shift which we consider as the noise power spectrum density. $f_{a}$ and $f_{b}$ are the integration limits that reduce the impact of the low and high frequencies noise. These limits were chosen here according to the expected signal frequency range. In the case shown in Figure 1, using a unique target, the zeroth moment is proportional to the square of the unscattered transmission ($M^{0}\propto e^{-2L(\mu_{a}+\mu_{s})}$). This proportionality is experimentally investigated at Section 4.1.

## 3. Materials and Methods

### 3.1. Skin Phantom Fabrication

In order to design proper skin phantoms, three types of materials are used: a scatter agent, an absorber, and a hardener ([12]). The scatter agent consists in microsphere particles (scatterers), the absorber is used for absorbing the light and the hardener is used to maintain the scattering agent and the absorber at fixed locations. By adjusting the concentration of these agents, it is possible to provide skin phantoms with controlled scattering, absorption and anisotropy coefficients.

The protocol used for phantoms fabrication has been described in [12,13]. Polydimethylsiloxane (PDMS) (Sylgard 184 Silicone Elastomer DOW/Corning) has been used as the base material (hardener) for skin phantoms. This product is a two-component product (silicone elastomer and curing agent) with a refractive index of 1.41 (close to skin tissue refractive index) that has been cured at room temperature during 24 h. We have used titanium (IV) oxide powder (TiO${}_{2}$) (13463-67-7, Sigma-Aldrich, St. Louis, MO 63178, USA) as the scatter agent [12]. In order to provide phantoms with a wide range of scattering coefficients, different concentrations of titanium oxide were used from 0.8 to 8 mg/mL with steps of 0.8 mg/mL. As the absorbing agent, a 0.2 mg/mL alcohol-soluble nigrosin (8005-03-6, Sigma-Aldrich) was added in all phantoms [13].

We first mixed titanium oxide with the silicone elastomer using an agitator for 30 min. Then nigrosin was mixed with the curing agent for 30 min. Afterwards the curing agent solution was added with a ratio of 1 to 10 to the silicone and mixed for 15 min. Then the mixture was placed in a chamber with a vacuum pump for about 30 min to remove air bubbles. Later, the liquid phantom was poured between two microscope slides separated by a spacer (which also acts as mold isolating the liquid mixture to remain between slides) with the desired thickness (0.5 mm). For one of the titanium oxide concentrations (4 mg/mL), various phantom thicknesses (0.1 to 0.5 mm) have been fabricated to further investigate the sensing depth estimation described in Section 4.3. Finally the mold was left at room temperature for 24 h.

### 3.2. Skin Phantom Characterization

To estimate the scattering and absorption coefficients of phantoms, a characterization setup has been developed. It is based on an integrating sphere (IS) (IS236A-4, Thorlabs Inc., Newton, NJ, USA) as shown in Figure 2. Results have been computed from Kubelka–Munk model equations [8,14,15]. As the light source, we have used a vertical-cavity surface-emitting laser diode (VCSEL-780, Thorlabs) producing a maximum power of 1.65 mW at the wavelength of 780 nm. This type of laser has been chosen because of its power stability in time, and its circular beam spot which illuminates uniformly skin phantoms samples.

The total transmission coefficient and diffuse reflection coefficients of the skin phantoms have been measured using the IS as shown in Figure 2. The collimated transmission was measured using the IS with an iris opened at 1 mm diameter just in front of the entrance port to avoid scattered photons entering into the sphere/detector (see Figure 2c). To reduce the probability of the scattered light presence in the IS, it was set 50 cm away from the sample. The measured scattering coefficients of the characterized phantoms are shown in Table 1.

Figure 3 shows the measured scattering coefficients versus titanium oxide concentration. The average deviation of the scattering coefficients from the linear fitting was about 5.2%, thereby, we can conclude that these results are in good agreement with the Mie theory and the Beer–Lambert law as demonstrated in [12,13], in which the scattering coefficient should be proportional to the scatter’s concentration.

## 4. Results

### 4.1. Zeroth Moment Proportionality with Unscatterred Transmission

As demonstrated in Section 2, the zeroth moment signal power is proportional to the unscatterred transmission coefficient. In order to experimentally verify this theory on our samples, we have set-up a standard OFI velocimeter system where a laser diode is pointing on a rotating metallic disk as depicted in Figure 4. A skin phantom was placed between the lens and the rotating disk.

The laser source is an AlGaAs Fabry–Perot laser, Thorlabs Inc. Newton, New Jersey, United States diode emitting at 785 nm with a maximum power of 50 mW. The beam is collimated on the surface of a rotating disk with a 8 mm focal length aspheric lens.

The signal of the monitoring photodiode, included in the laser package, is amplified and acquired by a digital oscilloscope with a sampling rate of 5 MHz and each acquisition consists in 1 million samples.

The signal total power is obtained by computing the Fast Fourier Transform (FFT) and integrating the power spectral density over the frequency limits ($f_{a}$, $f_{b}$) as described in (Equation 8) that were chosen to be $f_{a}=100$ Hz and $f_{b}=2.5$ MHz.

Figure 5 shows the evolution of the measured zero moment against the square of the unscattered transmission for all phantoms. Results from Table 1 were used to calculate the square of the unscattered transmission for each phantom using (Equation 3).

The average deviation from the linear relationship is about 6%. This deviation may be due to the inaccuracy in measuring the unscattered transmission and inhomogeneity of phantoms which is about 4 (±0.1) as given in Table 1. These results show that the proposed configuration can be used to obtain the unscattered transmission of a scattering media with reasonable accuracy.

This new ability of the OFI system to determine the unscattered trasnmission of a semi transparent media has been more developed in a patent [16].

### 4.2. Characterization of the 2D Resolution

Because the skin is a complex media, we have evaluated the impact of skin phantoms with the different optical properties depicted at Section 2 on the measurement spatial resolution. In that perspective, a microfluidic chip with a serpentine shape with variable channel spacing has been designed. The skin phantoms are positioned above the chip and the spatial resolution of our device is evaluated in its ability to detect the flow while scanning the chip. In Section 4.2.2, a Zemax simulated model of our sensor have been realised and tested. In Section 4.2.3, experimental results are exposed and commented.

#### 4.2.1. Microfluidic Chip Description

The experimental setup is composed by an OFI sensor mounted on a 2D mechanical scanning system and a custom microfluidic chip with a serpentine shape. Figure 6 shows the serpentine pattern. The spacing between each channel branches is progressive (10, 20, 50, 100, 200, 300, 400, 600, 700, 800 and 1400 µm). The channels have a square cross section with a side length of 100 µm. Scatterers in the channel were obtained by dilution of milk (1 part of milk and 10 parts of distilled water in weight). The optical source (2 mm circular source, AlGaAs Fabry–Perot laser diode emitting at 785 nm), is focused with a 8 mm focal lens into the channels plane, located under 0.5 mm thick of skin phantom. The angle between the optical axis and the normal to the microreactor plane is 10 degree. The OFI sensor is translated by steps of 20 µm and the zero moment of the signal spectrum is computed at each position.

#### 4.2.2. Simulated 2D Resolution Characterization

We simulated the power reflected by the capillaries with the software OpticsStudio Zemax (LLC 10230 NE Points Drive, Suite 500 Kirkland, Washington 98033 USA). The skin is modeled as a scattering “bulk” volume following the Henyey–Greenstein model. As regarding the mean paths, we used the inverse of the calculated scattering coefficient of Table 1. As regarding the anisotropy factor *g*, which represents the material’s directional dependence of a physical property, we simulated different factor: 0.95, 0.8 and 0.5 as this factor depends on the size of the particles included in the media [17]. The capillaries are following an “angle scattering” model, with a 180° angle, meaning they scatter in all directions in an isotropic way. The scattering coefficient of milk can highly vary depending on the milk itself from 20 to 100 mm^−1^. We chose the centered value of 60 mm^−1^, so a coefficient of 6 mm^−1^ when the milk is diluted by a factor 10, leading to a free path of 0.167 mm. For each simulations 10^5^ rays are generated, the maximum intersection per ray is equal to 4000 and the maximum segments per ray is set to 5×10^4^.

Figure 7, Figure 8 and Figure 9 present the simulations results for our system for three values of *g*. First we can see that the value of *g* strongly impacts the power of the back reflected light. With a high value of *g* and with a scanning step of 20 µm no matter the characteristic of the phantom, the channels with a spacing ranging from 100 µm to 1400 µm remain clearly visible. In the opposite way, the more the value of *g* decreases the more back reflected power does too. From these simulations, we can conclude that not only absorption and scattering characteristics of the skin are important parameters but also its anisotropy must be taken in consideration in order to evaluate the resolution of our device.

#### 4.2.3. Experimental 2D Resolution Characterization

The OFI sensors is the same that was depicted in Section 4.1. The sensor optical axis realizes an angle of 80 degrees with the flow direction. The photodiode signal was acquired using a National Instruments data acquisition card (BNC-2110) with a maximum sampling rate of 1 MHz. Figure 10 represents the scanning set up configuration.

To evaluate the limits in term of 2D resolution, we have performed several scans of the same region for the different skin phantom samples. Figure 11 represents the scanning results computed with the zeroth moment according to the skin phantom sample number. In the no phantom case, all channels are clearly visible. We can do the same observation for the samples 2 to 8. In addition, if we compare these results with simulation results of Figure 10 we can estimate that the anisotropy skin phantoms factor is around 0.9 which is in accordance with the theoretical analysis of light scattering properties of the titanium dioxide particle [17]. In the case of a skin phantom of 0.5 mm depth and a scattering coefficient of 13.2 mm^−1^ that correspond to the sample number 10, the sensor shows its limits because no channel is clearly visible.

### 4.3. Sensing Depth Estimation

The laser and the optical configuration were the same as the ones used in the previous section. The laser beam was focused onto a cylindrical polydimethylsiloxane (PDMS) fluidic chip consisting in a unique circular channel with a diameter of 320 µm and the skin phantom sample 5 with different thicknesses from 100 to 1000 µm by steps of 100 µm was placed over the PDMS fluidic chip. The channel was once again fed with diluted milk (1 part of milk and 10 parts of distilled water) at flow rates of 10, 30, 50 µL/min.

Figure 12 shows the zero moment of the signal spectrum against the skin phantom thickness for all three flow rates. As seen, the zero moment reaches to about the noise level at the thickness of 600 µm for all flow rates. Therefore, for the skin phantom number 5, the maximum sensing depth at which the Doppler shift can be observed in the spectrum of OFI signal is about 600 µm for the OFI sensor configuration used.

### 4.4. In Vivo Blood Vessel Imaging

In order to demonstrate the feasibility of the vein blood flow in vivo detection a transverse scanning of a superficial hand vein is performed. The scan length is set to 1.125 cm with a step size of 75 µm, thus producing a line of 150 pixels. For each pixel 61,440 points are acquired at a sampling frequency of 1 MHz. Figure 13 shows the computed zero moments that were computed using a restricted range of frequencies $f_{a}$, $f_{b}$ of, respectively 2 kHz and 22 kHz to limit the impact of the fringe induced by the natural motion of the forearm and taking into account the low velocity of the blood in the vein. The laser wavelength is 1310 nm.

Because the subcutaneous vein shape is almost circular, the parabolic profile observed is a strong indication of the blood flow detection. However, in order to eliminate the possibility of others origins of the increase of the zeroth moment such as the forearm motion induced fringes in the OFI signal, a spectral analysis has been conducted. The spectra of some representative pixels are plotted in Figure 13b). A continuous distribution of the Power Spectral Density, in the range of [2–22 kHz] is a clear mark of the presence of the Doppler phenomenon. The fact that the pixel with the higher moment value correspond to the most extended Doppler shifts in the spectra undeniably validates the measurement method and the interpretation that can be made of the parabolic shape of the moment evolution.

### 4.5. Dicsussion

In this paper, we have demonstrated the possibility to use the OFI technique in order to image blood flow beneath the skin. This ability to map the perfusion from a larger area by scanning the laser beam over the area of interest class our sensor into the Laser Doppler Perfusion Imager (LDPI) systems category [18] In contrast to other LDPI that use one component for the emission (Laser diode) and one component for the reception of the backscattered light (CMOS sensors) [19,20], our sensor uses only one component for light emission and reception. Thus, the main advantage of OFI sensors is the use of the packaged laser light as the light source, the interfering medium and the coherent detector all in one making the set up extremely compact, economic, self aligned and efficient, while keeping it contactless as classical LDPI sensors.

The results that we present in this study are showing very good accordance with the theoretical models as long as the measurement is performed with in vitro flows. As can be observed in Section 4.3, in vivo measurements lead to more complex observations. The impact of the displacement of the target (the patient skin) with regards to the sensor that produces interferometric fringes with high amplitude and large frequency range will inevitably impact the sensor signal. Regarding this aspect, the presented sensor, shows the same limits than classical Doppler perfusion imaging systems. In order to face this problem, different solutions have been investigated in the literature. Karlson et al. [21] shows that LDPI systems are sensitive to tissue motion not related to blood cells. By using a polarization filter.They demonstrated the fact that the polarized LDPI technique reduces movement artifacts. Bahadori et al. [22] have introduced an adhesive opaque patch (AOP) in order to substrate the Laser speckle contrast imager signal from the AOP from the laser speckle skin signal. Both methods have in common to add a new component in their sensors which increase the system complexity and cost. In this paper, motion artifacts cancellation has been done by using a dedicated signal processing which eliminates a large part of these undesirable signal perturbations by applying a sharp frequency selection for the computation of the moment. Nevertheless, we had to perform the measurement in the most possible static conditions for the forearm on which the measurement was performed. In real-life situations a solution will have to be found in order to isolate the Doppler induced signals from the ones induced by other mechanical movement.

When measuring blood skin blood flow perfusion with LDPI methods, it was found that there is a residual signal from the tissue called biological zero (BZ) which is present even where there is no flow [23]. The origin of this artifact is induced by the Brownian motion and movement of red blood cells. Measurements on excised tissue [23] and cadaver skin [24] have demonstrated that the BZ signal should be subtracted from the normal perfusion and especially in the case of very low measurements [25,26]. Kernick et al. [27], have developed a BZ-based flow model on in vivo and in vitro measurements. They demonstrated that the BZ varies linearly with temperature which is a characteristic of Brownian motion. This signal artifact has not been studied in this paper but must be taken in consideration for future work. We can plan to do a calibration procedure in order to evaluate the BZ noise for different temperatures. In addition, this experiment can be very interesting because the behavior of OFI sensor is also dependent of the operational temperature [28].

Another aspect that would require attention is the laser wavelength of the sensor. Wavelengths between (500 and 650 nm) are more absorbed by the skin (see [29]). This problematic have already been considered by others authors [30,31] for classical LDPI sensors. In the case of our OFI system, this absorption phenomenon will impact the penetration depth resolution of our sensor because the light feedback into the laser cavity will be reduced. In addition and a more problematic aspect, the use of this range of wavelength can damage irremediably the patient skin, causing skin burn or even skin cancer. The laser power and laser spot size on the skin surface must also be taken in consideration for skin safety. With a very small laser spot, the OFI system can reach a very good resolution, but the laser energy is then concentrated which enhance the risk of skin damage. A trade-off between resolution and skin health must be taken in consideration.

Eventually, the patient skin color shall be taken into consideration. in vitro results show that the more the absorption coefficient of the skin is important as less light will be fed back in the laser. In the frame of this work we have performed measurement exclusively on a Caucasian type skin and we believe that further developments of the in vivo application of this work shall evaluate the performances of our system on colored skins. Another parameter which will be interesting to take in consideration for in vivo measurements is the skin color. Indeed in vitro results shown that the more the absorption coefficient of the skin is important, less light feedback we have. In that perspective, it can be interesting to test our system on colored skin in order to evaluate the impact on this parameter on our system.

## 5. Conclusions

The impact of scattering media, similar to the human skin, set in the optical path of an optical feedback interferometry sensor has been investigated in both theoretical and experimental approaches. An experimental validation of the theoretical analysis shows that the zero order moment of the sensor signal spectrum is proportional to the unscatterred transmission coefficient. A Zemax simulation model of the measured system that includes a microfluidic chip and the skin phantom have been computed in order to better understand the impact of the skin anisotropy factor and its thickness. A series of skin tissues optical phantoms has been fabricated and characterized experimentally to validate the influence of skin-like scattering media in the OFI performance. Two parameters have been evaluated in particular: the spatial resolution in the skin plane and the maximum achievable sensing depth of a microscale channel flow buried in human skin. As a final and major achievement of this work, a demonstration of the feasibility of in vivo detection of a vein using the optical feedback interferometry sensing scheme. The experimental set-up that has been designed produces an image of the vein, namely to reveal the vein blood flow.

By the way, thanks to this device, the possibility to characterize subcutaneous veins shape can also be considered. The perspective is to realize a sensor which is able to take an image of the subcutaneous flow and, at the same time, to indicate relevant flow characteristics that can be used for a diagnostic purpose such as in vivo myography or blood clot detection. The possibility to design such a low cost, non-invasive, compact and easy-to-use device that can study the tissues microvascularization is the future challenge.

## Figures and Tables

**Figure 1 sensors-21-01300-f001:**
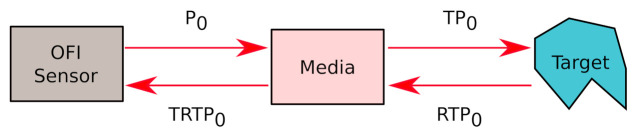
Schematic configuration of a laser under feedback with an attenuating medium placed between the laser and the target.

**Figure 2 sensors-21-01300-f002:**
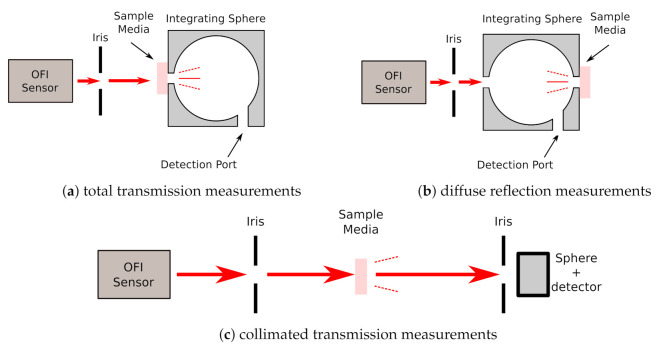
Skin phantoms characterization setup for total transmission measurements, diffuse reflection measurements and collimated transmission measurements.

**Figure 3 sensors-21-01300-f003:**
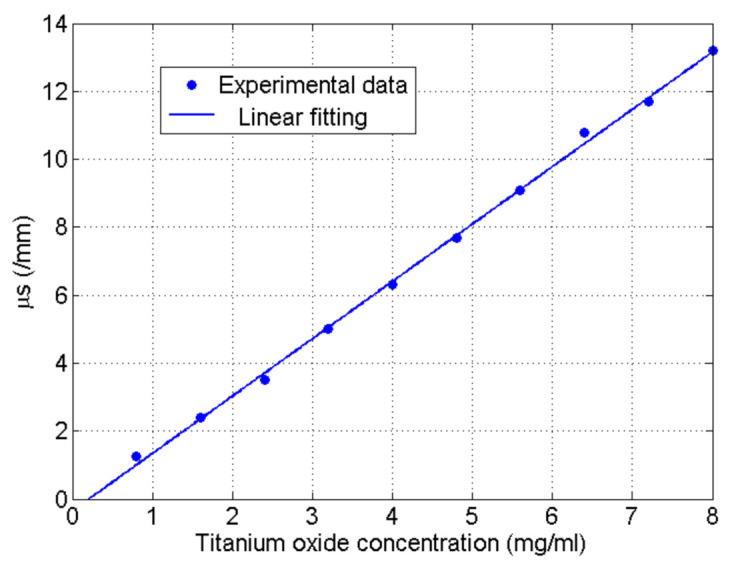
Measured scattering coefficient of the phantoms versus titanium oxide concentration

**Figure 4 sensors-21-01300-f004:**
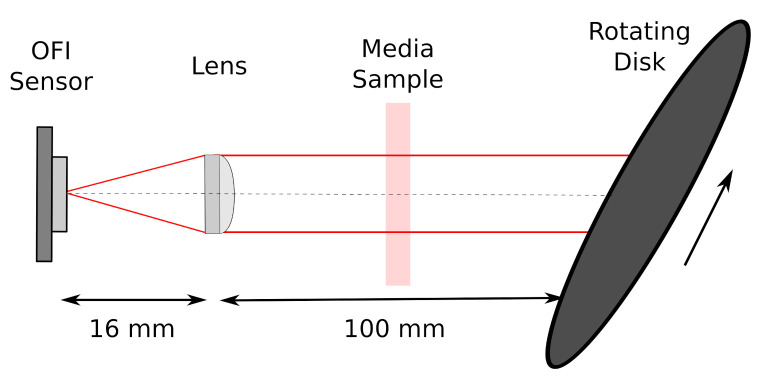
Schematic diagram of the velocity measurement with the skin phantom placed between the lens and the target which is a rotating disk.

**Figure 5 sensors-21-01300-f005:**
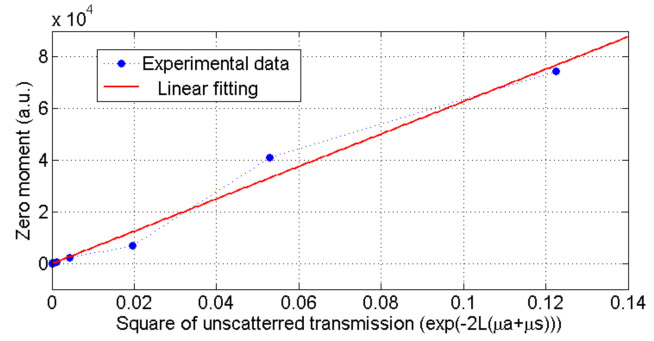
Zero moment versus square of unscattered transmission for fabricated phantoms.

**Figure 6 sensors-21-01300-f006:**
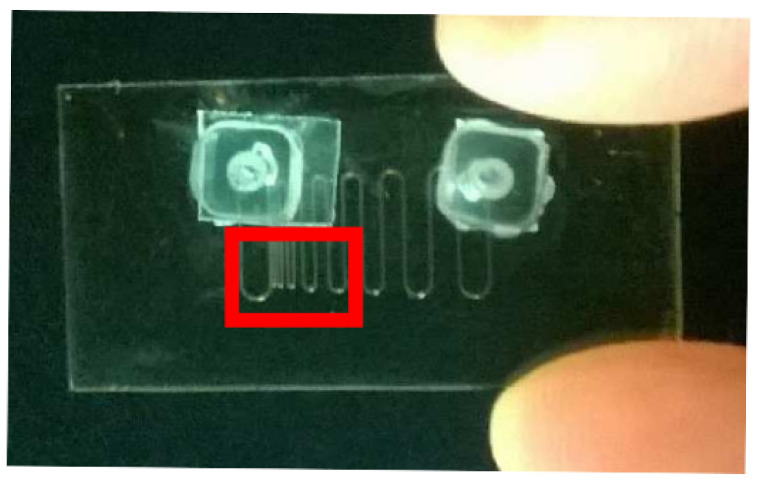
Photography of the microfluidic chip with the serpentine 100 µm square section channels and spacing ranging from 10 to 1400 µm. The red rectangular describes the actual scanning area.

**Figure 7 sensors-21-01300-f007:**
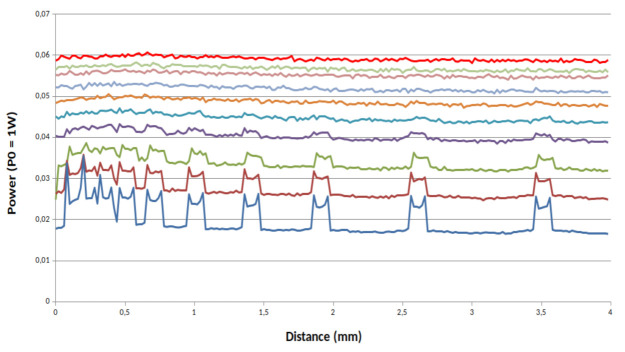
Power reflected by the channels full of milk for a phantom with a total transmission for *g* = 0.5: from top solid line (red solid line sample 1) to bottom blue solid (sample 10).

**Figure 8 sensors-21-01300-f008:**
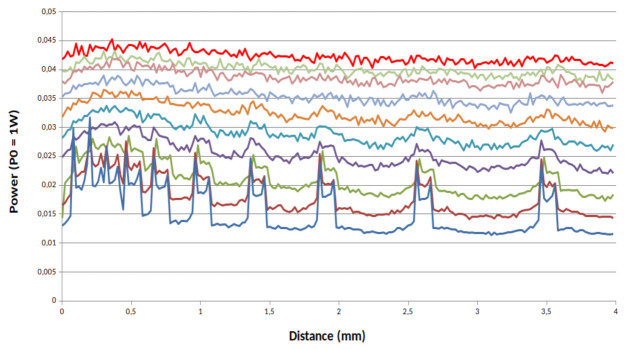
Power reflected by the channels full of milk for a phantom with a total transmission for *g* = 0.8: from top solid line (red solid line sample 1) to bottom blue solid (sample 10).

**Figure 9 sensors-21-01300-f009:**
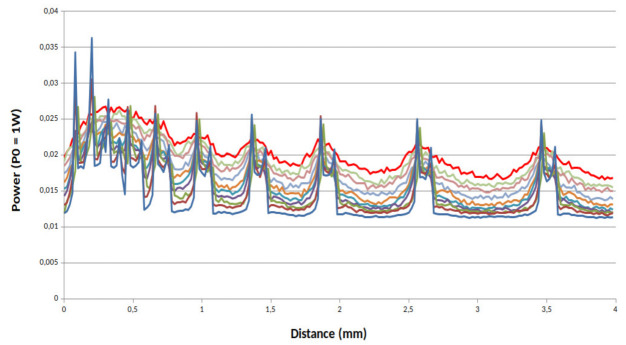
Power reflected by the channels full of milk for a phantom with a total transmission for *g* = 0.95: from top solid line (red solid line sample 1) to bottom blue solid (sample 10).

**Figure 10 sensors-21-01300-f010:**
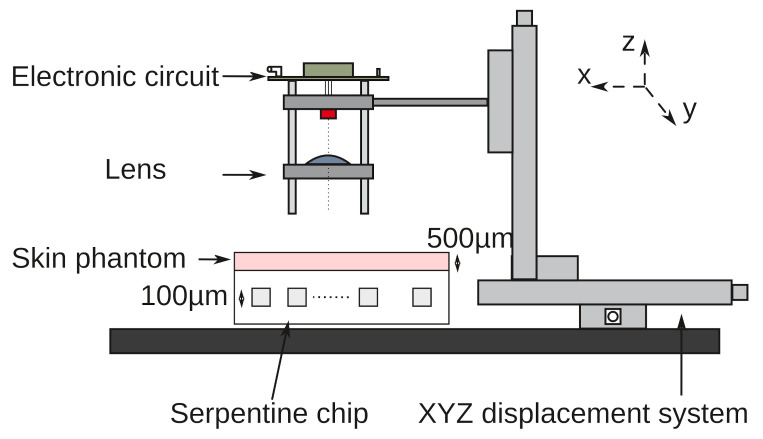
Experimental set up for OFI (Optical Feedback Interferometry) 2D resolution characterization.

**Figure 11 sensors-21-01300-f011:**
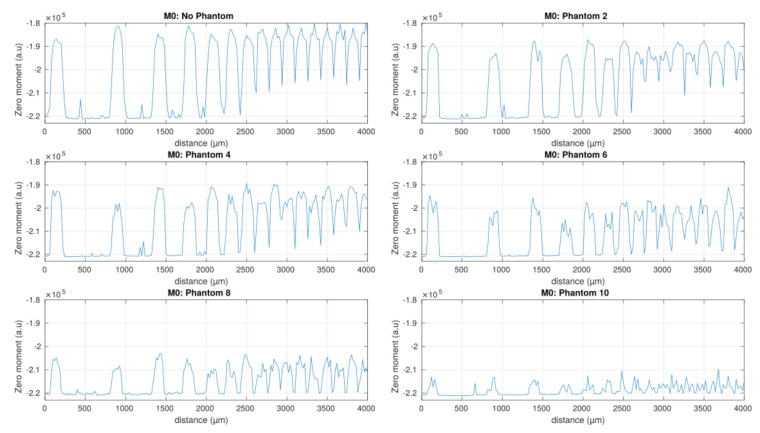
Experimentalresults for OFI 2D resolution.

**Figure 12 sensors-21-01300-f012:**
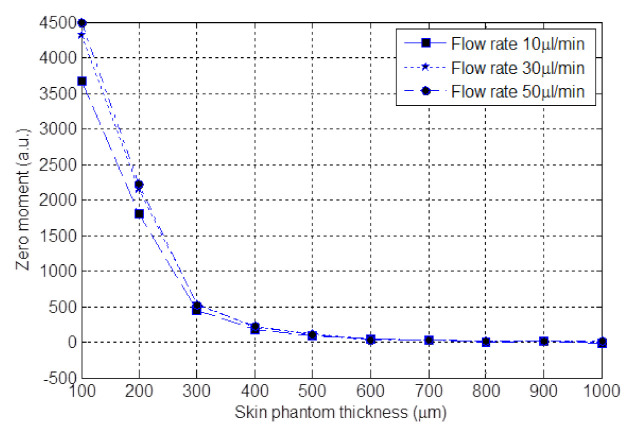
Zero moment obtained for skin phantom number 5 with different thickness at the rates 10, 30, 50 µL/min.

**Figure 13 sensors-21-01300-f013:**
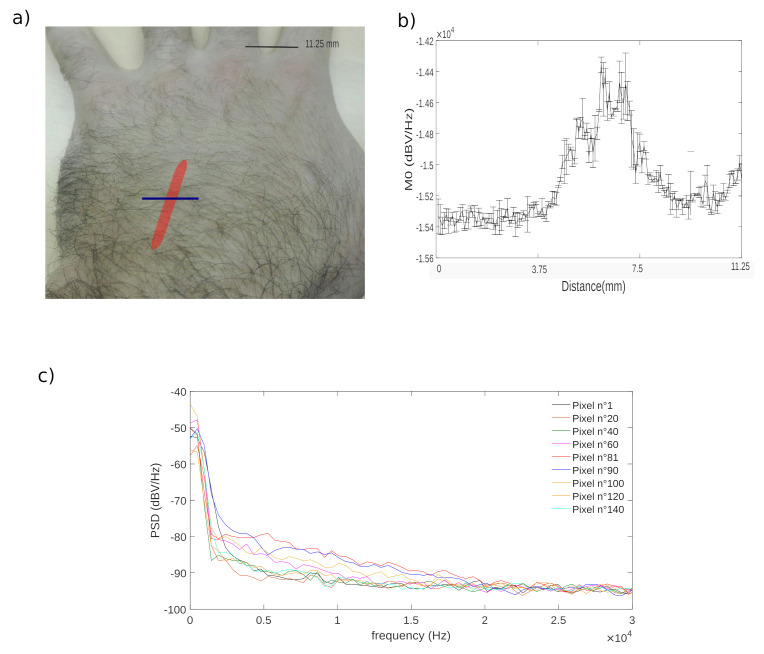
(**a**) Photography of the scanning zone (the red thick line represents the position of the vein, the blue thin line represents the transverse scanning line), (**b**) evolution of the zeroth moment and (**c**) spectra of different pixels along the scan line transverse to the superficial vein).

**Table 1 sensors-21-01300-t001:** Measured optical properties of fabricated phantoms.

Phantom Sample	TiO${}_{2}$ Concentration (mg/mL)	Scattering Coefficient (mm${}^{-1}$) (±0.1)
1	0.8	1.2
2	1.6	2.4
3	2.4	3.5
4	3.2	5.0
5	4.0	6.3
6	4.8	7.7
7	5.6	9.1
8	6.4	10.8
9	7.2	11.7
10	8.0	13.2

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
