# Peer review of "Methods and Limits for Micro Scale Blood Vessel Flow Imaging in Scattering Media by Optical Feedback Interferometry: Application to Human Skin"

_sensors, 2021, doi:10.3390/s21041300_

Round 1

Reviewer 1 Report

In this study, optical feedback interferometry (OFI) was applied as a sensor for blood flow imaging of skin vascularization. The OFI based flow sensing was analyzed in different skin-tissue optical phantoms and human skin. A novel method for blood perfusion computation using zeroth moment was proposed and investigated.

Comments

The work subject is interesting and with possible skin vascularization monitoring application, requiring minor changes to be considered for Sensors publication. The modifications below are suggested in the final version:

  • Equation 3, is the correct form Rt, or Rt?
  • Table 1, scattering coefficient (mm-1) (+/- 2). It is necessary to present the rounding of the uncertainty properly, for example: (1.25 +/- 2) should be (1 +/- 2).
  • The spectra of some representative pixels plotted in Figure 13 (b), could be presented in unit length? In Figure 13 (a) the blue line could be estimated in units of length.
  • In vivo blood vessel imaging. Do the results shown to depend on the laser wavelength or power of the OFI sensing used and the color of the skin investigated?

Author Response

Dear reviewer,

Please find our comments in the attachment letter.

Best regards

Adam QUOTB

Reviewer 2 Report

Thank you for giving me this opportunity to review this research article entitled, "Methods and limits for micro scale blood vessel flow imaging in scattering media by Optical Feedback Interferometry: Application to human skin".

I here carefully reviewed the submitted set of the manuscript and found the contents of this research conducted possibly merits of publication. However, the style of the submitted set of the article should be really awkward in the present form.

 Then, once again, I confirmed the "Instructions for authors", which describes that "The abstract should be a single paragraph and should follow the style of structured abstracts, but without headings: 1) Background: Place the question addressed in a broad context and highlight the purpose of the study; 2) Methods: Describe briefly the main methods or treatments applied. Include any relevant preregistration numbers, and species and strains of any animals used. 3) Results: Summarize the article's main findings; and 4) Conclusion: Indicate the main conclusions or interpretations. The abstract should be an objective representation of the article: it must not contain results which are not presented and substantiated in the main text and should not exaggerate the main conclusions. This Abstract doesn't conform to it.

Also, I've never read and reviewed the article without the Discussion section or discussion in detail considering and comparing the relevant researches and the publications.

If the Discussion section is to be excluded, the Results section should be combined with Discussions referring to the recent publications and the relevant research articles. The author instructions also describe that "Authors should discuss the results and how they can be interpreted in perspective of previous studies and of the working hypotheses. The findings and their implications should be discussed in the broadest context possible and limitations of the work highlighted. Future research directions may also be mentioned. "

I'm very sorry but this submitted article must be substantially revised and resubmitted for re-evaluation and re-reviewing for suitability for publication.  

Author Response

Dear reviewer,

Please find our comments in the attachements.

Best regards

Adam QUOTB

Round 2

Reviewer 2 Report

Thanks for giving me this opportunity to re-review the revised manuscript.

I here carefully re-reviewed the revised manuscript and was so disappointed at the authors' revisions without inappropriate corrections. At this current form, I can't support publication of this article I'm deeply afraid, with only describing and introducing the authors' experiments and results.

Once again, let me ask you to conform to the standard of scientific publication and the journal guideline.

"The abstract should be a single paragraph and should follow the style of structured abstracts, but without headings: 1) Background: Place the question addressed in a broad context and highlight the purpose of the study; 2) Methods: Describe briefly the main methods or treatments applied. Include any relevant preregistration numbers, and species and strains of any animals used. 3) Results: Summarize the article's main findings; and 4) Conclusion: Indicate the main conclusions or interpretations. The abstract should be an objective representation of the article: it must not contain results which are not presented and substantiated in the main text and should not exaggerate the main conclusions. This Abstract doesn't conform to it.

Also, I've never read and reviewed the article without the Discussion section in details referring to, considering and comparing the relevant researches and the relevant publications. They have just described repetitions of the Results. 

If the Discussion section is to be excluded, the Results section should be combined with Discussions referring to the recent publications and the relevant research articles. The author instructions also describe that "Authors should discuss the results and how they can be interpreted in perspective of previous studies and of the working hypotheses referring to the previous researches and the recent publications. The findings and their implications should be discussed in the broadest context possible and limitations of the work highlighted. Future research directions may also be mentioned. "

Author Response

Dear reviewer,

please find in the the attachment our comments.

Best regards
